# Antifungal and Anti-Inflammatory Activities of PS1-2 Peptide against Fluconazole-Resistant *Candida albicans*

**DOI:** 10.3390/antibiotics11121779

**Published:** 2022-12-08

**Authors:** Jong-Kook Lee, Soyoung Park, Young-Min Kim, Taeuk Guk, Jong Kwon Choi, Jin-Young Kim, Min-Young Lee, Mi-Kyeong Jang, Seong-Cheol Park

**Affiliations:** 1Department of Chemical Engineering, Suncheon National University, Suncheon 57922, Republic of Korea; 2Department of Clinical Laboratory Science, Daejeon Health Institute of Technology, Daejeon 34504, Republic of Korea

**Keywords:** antimicrobial peptides, *Candida albicans*, anti-inflammatory effect, drug-resistance, cytokines

## Abstract

Clinically, fungal pneumonia rarely occurs in adults, and invasive fungal infections can cause substantial morbidity, and mortality due to sepsis and septic shock. In the present study, we have designed peptides that exhibit potent antifungal activities against fluconazole-resistant *Candida albicans* in physiological monovalent, and divalent ionic buffers, with minimal fungicidal concentrations ranging from 16 to 32 µM. None of these tested peptides resulted in the development of drug resistance similar to fluconazole. Among them, the PS1-2 peptide did not induce stimulation of macrophages by *C. albicans,* and it exerted antifungal and anti-inflammatory effects against *C. albicans*-induced intratracheal infection, in an acute lung injury mouse model. PS1-2 is likely a novel therapeutic agent for the control, and prevention of drug-resistant *C. albicans* infection, and our findings may be useful for designing antimicrobial peptides to combat fungal infection.

## 1. Introduction

The emergence of new drug-resistant pathogens and the rapid increase in the number of patients infected by these pathogens, are currently the serious issues in human health worldwide, which can also be considered a risk factor that can cause another pandemic after COVID-19 [1]. Pathogenic fungi, though less studied, are rapidly becoming resistant to antifungal agents because of treatment with sub-therapeutic doses or misuse of antifungal drugs, nosocomial transmission, and large-scale use of fungicides in agriculture [1,2,3,4]. Unlike bacteria, the development of fungal drug resistance is slow globally, because drug-resistance genes for antifungal agents must be independently generated [1,5]. Current pharmacological options available for fungal infections, include azoles (fluconazole), echinocandins (caspofungin), and polyenes (Amphotericin B). However, they have to be administered at low doses due to their cytotoxic effects, which leads to reduced antifungal efficacy; resistant fungi have been isolated from many such patients [6,7,8]. The Centers for Disease Control and Prevention (CDC) has estimated a need to hospitalize 75,000 patients per year with drug-resistant fungal infections and additionally fungal infections have increased in the wake of the COVID-19 pandemic [9,10].

The most widespread human fungal pathogen, *Candida albicans*, causes both superficial and systemic infections, colonizing multiple mucosal sites, including the oral cavity, and gastrointestinal and urogenital tracts, in symptomatically immunocompromised patients, but not in healthy individuals [11]. In particular, *C. albicans* can change its morphology from a commensal budding yeast to a pathogenic filamentous form, with the ability to damage host cells by triggering adhesion and invasion on the cell surface [12]. If local infections are not treated immediately or adequate treatment is not provided, they can disseminate through the bloodstream, resulting in systemic infections with high mortality [13].

Macrophages and neutrophils are the key phagocytes of primary immunological defense in mammals against pathogenic fungi; however, impaired phagocytosis is a major risk factor for exacerbating fungal infection severity [14,15]. The recognition of pathogen-associated molecular patterns, that results in accumulation of phagocytes at the site where the fungal cells are located, is important for the phagocytic clearance of pathogenic fungi [15]. This recognition is mainly stimulated by unmasked β-1,3-glucan, a proinflammatory polysaccharide that can induce release of inflammatory cytokines from macrophages. Engulfed fungi are killed or removed by the production of reactive nitrogen and oxygen species in the phagolysosome [15,16,17]. However, when an inflammatory response is excessively induced, cytokines are continuously overexpressed and can move to the whole body through blood vessels, resulted in sepsis [16]. Therefore, there is a need to develop new drugs to tackle fungal pathogens and prevent excessive inflammatory responses.

Antimicrobial peptides (AMPs) are promising candidates that can limit drug-resistant *Candida* cells via growth inhibition, or candidacidal action [18,19]. AMPs are well-known host defense molecules in all living organisms, including microbes, vertebrates, and invertebrate species [20,21], and possess common properties, such as having 12–50 amino acids, cationic and amphipathic nature, and a broad spectrum of activity [20,21]. In addition, their antimicrobial activity is achieved either by membrane destabilization, such as pore formation and membrane disruption, or metabolic inhibition in the cytosol through cell penetration [21,22]. They have different mechanism of action from conventional antibiotics, can avoid drug resistance, and are being considered as novel alternatives to synthetic antibiotics.

We previously reported that PS1 peptides, a novel series of peptides with repeated sequences of ‘XWZX’ sequence motif (X: lysine or arginine, Z: leucine, tyrosine, valine, or glycine), have α-helical structure. They exert potent antibacterial effects via membranolytic action [23] and anti-biofilm action via inhibition of biofilm formation, and disruption of preformed biofilms in *Pseudomonas aeruginosa* and *Staphylococcus aureus* [24]. In particular, the PS1-3 peptide inhibits *C. albicans* biofilm formation, and reduction at the minimum fungicidal concentration on the contact lens surface [25].

The present study focused on the in vitro and in vivo antifungal, and anti-inflammatory activities of PS1-2 ((KWYK)_3_) against fluconazole-resistant *C. albicans*. We investigated the fungicidal activity, fungal cell-binding affinity, time-killing kinetics, induction of drug resistance, and inhibition of macrophage stimulation in vitro. Furthermore, the in vivo antifungal and anti-inflammatory effects of PS1-2 were demonstrated in a lung injury mouse model.

## 2. Results and Discussion

### 2.1. In Vitro Antifungal Activity of PS1-2, PS1-5 and PS1-6 against Drug-Resistant C. albicans

PS1 peptides exert antibacterial and anti-biofilm actions against drug-sensitive, and drug-resistant bacterial pathogens in both, in vitro and in vivo assays [23,24]. Among them, PS1-3 exhibited potent biofilm-inhibiting, and -eliminating activities against fungal biofilms on contact lenses [25]. To select the peptides with excellent antifungal and anti-inflammatory effects as lead candidates, the present study was carried out using PS1-2 ((KWYK)_3_), PS1-5 ((RWYR)_3_), and PS1-6 ((KWLK)_3_) peptides.

*C. albicans* is a common commensal fungus that colonizes the human skin and inside the body, such as the mouth, gut, throat, and vagina [26]. It causes fungal infections in the bloodstream or internal organs including the kidneys, heart, and brain [26]. To apply antimicrobial peptides to clinical therapy, it is important to maintain their antimicrobial activity under physiological conditions in human body fluids. The dental foci, gastric lumen, vagina, and lung-lining fluids in cystic fibrosis and asthma have an acidic environment [27,28,29,30]. The concentrations of K^+^, Ca^2+^, and Mg^2+^ ions, and salt concentrations are different in the tissues, cells, and fluids of each organ. The electrostatic attraction between antimicrobial peptides and pathogens can be interrupted by these peptides. The hyphal growth of *C. albicans* is promoted at temperatures above 35 °C [27,28]. This hyphal form plays an important role in the invasion of epithelial cells and tissue damage [31].

We determined antifungal activity of the peptides against drug-resistant *C. albicans* CCARM 14007, under different ionic concentrations, pH, and temperature. As shown in Table 1, PS1-2 was measured with an MFC value of 32 µM under all buffer conditions except CaCl_2_ (64 µM), indicating that it can exhibit antifungal activity under most physiological conditions in the human body. The fungicidal activity of PS1-5, in which lysine (K) is substituted with arginine (R), was enhanced in acidic buffers, but was reduced in monovalent and divalent ionic buffers. This indicated that its antifungal effect may decrease when it enters the body. MFC of PS1-6, in which tyrosine (Y) was substituted with leucine (L), was not affected by pH and increased to ion changes (Table 1). The pH provides little effect on the attachment of peptides to the *C. albicans* surface because it is a change in hydrogen ions. However, since the side chains of the peptides possess 6 hydrophilic cations, monovalent and divalent ions may reduce the binding efficiency of the peptides to *C. albicans*.

### 2.2. Fungicidal Kinetics of PS1-2, PS1-5 and PS1-6

To determine whether the antifungal action of the peptides is fungistatic, only inhibiting the growth of *C. albicans*, or fungicidal, that directly kills them, the killing kinetics of PS1-2, PS1-5, PS1-6, and fluconazole were evaluated against *C. albicans* at 1 min intervals for 5 min in SP (10 mM sodium phosphate buffer, pH 7.2) or PBS (pH 7.2). Alterations in the killing kinetic pattern under low salt (SP) and high salt (PBS) conditions can predict the interaction and mode of action of peptides on yeast. Because propidium (PI) dye can stain the live cells only temporarily, it will be released from the cytosol within minutes if the peptides cause cell death by damaging the cell membrane, resulting in the emission of red fluorescence by *C. albicans* cells. Flow cytometry results showed that PS1-2-treated *C. albicans* cells expressed maximum red fluorescence in SP or PBS within 1 or 4 min, and 86.2 and 87% of cells were killed after 5 min, respectively (Figure 1). PS1-5 killed 50% of the *C. albicans* cells in 1 min and thereafter remained stable for 5 min in both buffers. PS1-6 killed more than 90% of *C. albicans* cells at 1 min in PBS and 99.6% at 5 min in SP. However, fluconazole did not cause cell death at 128 µM since the *C. albicans* used is a fluconazole-resistant species (Figure 1). These results suggest that the antifungal activity of all tested peptides was fungicidal through membranolytic action. Although PS1-6 exhibits high antifungal activity as shown in Table 1 and Figure 1, previous studies have reported that it has significant toxicity [23]. Therefore, the subsequent experiments were performed with PS1-2.

### 2.3. Binding Affinity of FAM-Labeled PS1-2 Peptide to C. albicans Cells

To analyze the binding affinity of the PS1 peptide to the *C. albicans* membrane, PS1 peptides were 5-carboxyfluorescein (FAM)-labeled at the N-terminus. Since the FAM labeling on PS1-2 increases its hydrophobicity, this may affect its antifungal activity and mechanism against *C. albicans*. Therefore, we treated *C. albicans* with a mixture of FAM-labeled PS1-2 and free PS1-2 at a 1:9 molar ratio to minimize this effect. As shown in Figure 2a, the FAM-labeled PS1-2 peptides accumulated on the cell surface. In the presence of FAM-labeled PS1-2 peptide, the number of cells emitting green fluorescence increased to 41.16%, compared with cells in the absence of that (Figure 2b) and the plate well incubated with FAM-labeled PS1-2 peptide expressed strong green fluorescence (Figure 2c). As both these results and time-killing kinetic results, we suggest that the hydrophilic side chains of the PS1-2 peptide rapidly bind to the surface of *C. albicans* cells via electrostatic interaction, and it can rapidly kill them via membrane disruption.

### 2.4. Peptides Did Not Induce the Drug-Resistance

To apply AMPs in clinic, it should not induce drug-resistance, compared to conventional antibiotics. We measured each MFC against drug-sensitive *C. albicans* (KCTC 7270) to compare the degree and rate of drug-resistance between fluconazole and peptides. Figure 3 shows the fold-changes of MFC based on the first passage in *C. albicans* cells, which survived at sub-lethal concentrations of each passage. As shown in Figure 3, during six passages, the MFC of fluconazole significantly increased from 128 to 2048 µM at 28 °C, and from 128 to 4096 µM at 37 °C. We found that conventional antifungals rapidly induced drug resistance. However, peptides did not change its MFC at either temperature (Figure 3). The MFC of fluconazole increased exponentially over six passages, and none of the other peptides changed the MFC (data not shown). These results suggest that the membrane-permeating mechanism of peptides does not easily cause drug resistance.

### 2.5. Protection for Protease Degradation

To investigate the proteolytic stability of the peptides, the peptide solutions were pre-treated with trypsin for 40 min and added to *C. albicans* cells. After 24 h of incubation, cells were spread on agar plates (Figure 4). Although not all peptides were completely protected against trypsin proteolysis, *C. albicans* growth was inhibited > 70%, indicating partial protection. In particular, PS1-6 showed significant fungal growth inhibition compared to the others. We suggest that the increase in its hydrophobicity induces self-aggregation of the peptides, and these aggregates cannot be digested by trypsin because the cleavage sites of the peptides are not exposed. Although the antifungal activity of PS1-2 is partially inhibited by trypsin, we suggest that it inhibits the 70% growth of *C. albicans* in its MFC and it can completely inhibit the rest by the human immune system. In addition, since the cytotoxicity of PS1-2 is significantly lower than that of PS1-6 [23], it is expected to completely inhibit the growth of *C. albicans* at a dose higher than its MFC.

### 2.6. In Vitro Anti-Inflammatory Activity of PS1-2 Peptide in Co-Culture of RAW 264.7 and C. albicans

To investigate the effect of PS1-2 on *C. albicans*-induced inflammatory response, murine RAW 264.7 cell, a macrophage was used. Pre-cultured RAW 264.7 cells were incubated with *C. albicans* cells, in the absence or presence of fluconazole or PS1-2. As shown in Figure 5a, the major morphology of the cells was circular when the RAW 264.7 cells were cultured in the medium without any treatment. Lamellipodia extension and spreading of cells that are associated with macrophage activation, and phagocytosis were observed in the presence of *C. albicans* cells as well as in the presence of *C. albicans* cells treated with fluconazole. Interestingly, RAW 264.7, treated with *C. albicans* cells and PS1-2, showed circular forms, indicating that PS1-2 can protect against fungus-stimulated inflammation in macrophages. On the other hand, to clarify anti-inflammatory effects of PS1-2, the protein expression of Toll-like receptor (TLR)-2 (Figure 5b) and tumor necrosis factor (TNF)-α (Figure 5c) in macrophages was analyzed using their monoclonal antibody. TLR-2 expression was remarkably inhibited in the presence of *C. albicans* cells and PS1-2, but *C. albicans* cells with non-treatment stimulated TLR-2 up-regulation (Figure 5b). Overexpression of TNF-α, a pro-inflammatory cytokine, was observed in *C. albicans* cells without/with fluconazole; however, PS1-2 did not upregulate TNF-α in the presence or absence of *C. albicans* cells.

### 2.7. In Vivo Anti-Inflammatory Effects of PS1-2 in Acute Lung Injury

To assess the anti-inflammatory action of the peptides in acute lung injury, *C. albicans* CCARM 14007 cells were intratracheally instilled into the mouse lung. Nitric oxide (NO) and TNF-α secretion levels were quantified in whole blood serum (Figure 6a,b). Fluconazole treatment of *C. albicans*-infected mice did not inhibit NO production, but TNF-α secretion was inhibited by approximately 65%. Interestingly, PS1-2 instillation with/without *C. albicans* infection resulted in similar NO and TNF-α levels in blood serum in non-treated control mice, indicating the inhibitory effects of PS1-2 on *C. albicans*-induced inflammation via its fungicidal action.

In order to investigate the pathological effects of PS1-2, lung tissues excised from mice were stained with hematoxylin and eosin (H&E), alcian blue, or alcian blue and eosin. Alcian blue is a stain that is used to visualize acidic epithelial, and connective tissue mucins. Respiratory tracts overproduce mucus during acute injuries and in chronic conditions containing cystic fibrosis, bronchitis, and asthma [32,33]. As shown in Figure 6c, *C. albicans*-infected mice with fluconazole showed an increase in the release of mucins (red arrows), and inflammatory cell infiltrates around the bronchus and alveoli (blue arrows), indicating severe infection, as well as only fungal infection. However, treatment of infected mice with PS1-2 (2 mg/kg) displayed a significantly low level of inflammatory infiltrates and mucin release, and its treatment in non-infected mice did not induce inflammation, similar to the lung tissues of non-infected mice. These results reveal that the PS1-2 peptide can be used as an anti-fungal for *C. albicans* infection, although more preclinical studies are needed.

## 3. Materials and Methods

### 3.1. Materials

*N*,*N*′-Diisopropylcarbodiimide (DIC), 3,6-dioxa-1,8-octane-dithiol (DODT), trifluoroacetic acid (TFA), and triisopropylsilane (TIS) were purchased from Tokyo Chemical Industry Co., Ltd. (Tokyo, Japan). Ethyl 2-cyano-2-(hydroxyimino)acetate (Oxyma Pure), 9-fluorrenylmethoxycarbonyl (Fmoc) amino acids, and Rink Amide ProTide Resin were obtained from CEM Co. (Matthews, NC, USA). All other materials were of analytical grade and used as received. Propidium iodide (PI), Dulbecco’s modified Eagle medium (DMEM), Roswell Park Memorial Institute (RPMI)-1640, GlutaMax, fetal bovine serum (BSA), antibiotics (penicillin/streptomycin), and FAM were purchased from Thermo Fisher Scientific (Waltham, MA, USA). C. albicans CCARM 14007, which was first isolated in 1999, and was found to be resistant to amphotericin B, flucytisine, and fluconazole, was purchased from the Culture Collection of Antimicrobial-resistant Microbes (CCARM), in South Korea.

### 3.2. Synthesis of the PS1P by Fmoc Solid-Phase Method

PS1-2 (Lys-Trp-Tyr-Lys-Lys-Trp-Tyr-Lys-Lys-Trp-Tyr-Lys, (KWYK)_3_), PS1-5 (Arg-Trp-Tyr-Arg-Arg-Trp-Tyr-Arg-Arg-Trp-Tyr-Arg, (RWYR)_3_), and PS1-6 (Lys-Trp-Leu- Lys-Lys-Trp-Leu-Lys-Lys-Trp-Leu-Lys, (KWLK)_3_) peptides were synthesized on Rink Amide ProTide Resin (0.58 mmol/g). The peptides were synthesized using a Liberty microwave peptide synthesizer (CEM Co.). Default standard 90 °C coupling (five times the synthesis scale, Oxyma/DIC), and deprotection (20% piperidine in dimethylformamide (DMF)) methods were used. To generate N-terminal fluorescently labeled peptides, FAM was added to the peptide-bound resin. The synthesized peptides were cleaved in a cleavage cocktail (TFA/TIS/DODT/H_2_O, 92.5/2.5/2.5/2.5) for 40 min at 40 °C. Peptides were isolated by a Zorbax C_18_ column (300 Å, 7-μm, 21.2 × 250 mm) on a Shimadzu semi-preparative HPLC system (Tokyo, Japan), using a 10–70% acetonitrile gradient at 1 mL/min in water with 0.05% TFA. The molecular masses of the peptides were measured on a matrix-assisted laser desorption ionization (MALDI) mass spectrometer (Kratos Analytical, Manchester, UK).

### 3.3. Anti-Fungicidal Activity

*C. albicans* CCARM 14007 cells pre-grown in yeast extract-peptone-dextrose (YPD) medium were suspended in 10 mM SP buffer (pH 5.5, 6.0, or 7.2) containing 20% YPD medium or SP buffer supplemented with 150 mM NaCl, 24 mM KCl, 6 mM CaCl_2_, or 6 mM MgCl_2_ in the presence of 20% YPD medium (2 × 10^4^ cells/mL). The peptides or fluconazole were 2-fold serially diluted in 96-well plates with the indicated buffers, and the suspended cells were added to each well in equal volumes. After incubation for 24 h at 28 °C or 37 °C, 10 µL aliquots were removed from each well and spread onto YPD agar plates, followed by further incubation for 24 h. The lowest drug concentration at which no more than one colony was visible on the agar plate, was defined as a minimum fungicidal concentration (MFC).

### 3.4. Binding Affinity of FAM-Labeled Peptide

The pre-grown *C. albicans* CCARM 14007 cells were suspended in phosphate-buffered saline (PBS) containing 10% YPD media at a density of 5 × 10^5^ cells/mL. The cells with FAM-labeled peptides at MFC were incubated for 1 min at 28 °C and washed three times with PBS to remove free peptides. The cells were observed under a fluorescence microscope (OPTINIT KCS3-160S, Korea Lab Tech), fluorescence signal imaging system (FOBI, Cellgentek, Deajeon, South Korea), and flow cytometry analysis (Attune NxT acoustic focusing cytometer, ThermoFisher Scientific Co., Waltham, MA, USA).

### 3.5. Time-Killing Kinetic Assay

*C. albicans* CCARM 14007 cells grown to the mid-log phase at 28 °C were suspended in 10 mM sodium phosphate (pH 7.2) buffer or PBS (pH 7.2) buffer with 10% YPD media at a density of 5 × 10^5^ cells/mL. After 10 min pre-incubation of PI dye with the cell suspensions, peptides and fluconazole at the MFC were added, followed by incubation for 1, 2, 3, 4 and 5 min. The cells were analyzed by fluorescence-activated cell sorting (FACS) system (Attune NxT acoustic focusing cytometer).

### 3.6. Drug-Resistance Assay

To investigate the induction of drug resistance, *C. albicans* KCTC 7270 cells were prepared as for antifungal assay described above, at concentrations ranging from 1 to 128 μM. After 24 h incubation, MFCs were determined by the above method, and the surviving cells at half MFCs of each sample were treated with the diluted samples, repeating the process six times.

### 3.7. Trypsin Hydrolysis Assay

The peptides were incubated at 37 °C for 40 min with 1 µg/mL trypsin and added to suspensions of *C. albicans* CCARM 14007 (1 × 10^4^ cells/mL), followed by 24 h incubation. YPD agar plates spread with 10 µL of cell suspensions was incubated for 24 h at 28 °C. Negative and positive control are in the absence and presence of *C. albicans* cells without peptides, respectively.

### 3.8. Anti-Inflammatory Assay

RAW 264.7 cells were cultured in DMEM supplemented with 10% FBS, 1% penicillin/streptomycin, and GlutaMax at 5% CO_2_ at 37 °C. The RAW 264.7 cells (2 × 10^5^ cells/mL) suspended with RPMI-1640 medium were seeded into 12-well plates and incubated for 24 h. The attached cells were treated with *C. albicans* CCARM 14007 (1 × 10^5^ cells/mL) and fluconazole (128 µM), or PS1-2 peptide (32 µM) was added. After 24 h incubation, cells were washed and harvested in PBS. A total of 1000 µL of blocking solution (5% bovine serum albumin with PBS) was added to each sample, incubated at 25 °C for 30 min, removed by centrifugation at 1000× *g*, and then incubated with the primary antibody (anti-TLR2, or anti-TNF-α; Abcam, Cambridge, UK, 1:200 dilution) for 1 h. The cells were washed with PBS and incubated with a secondary antibody (fluorescein isothiocyanate-conjugated goat anti-mouse IgG, 1:500 dilution; Santa Cruz Biotechnology) for 1 h. Cells expressing TLR2 were measured using an Attune NxT acoustic focusing cytometer. TNF-α expression was visualized using a confocal laser-scanning microscope (CLSM, A1R HD 25, Nikon, Tokyo, Japan).

### 3.9. In Vivo Study

Animal study was approved by the Institutional Animal Care and Use Committee (IACUC) in Sunchon National University, South Korea (SCNU IACUC-2019-10). Six-week-old BALB/c mice (Koatech Co., Pyongtaek, Gyeonggido, Korea) were randomly divided into five groups (4 mice per group): (1) control (100 µL of PBS); (2) *C. albicans* CCARM 14007 (100 µL of 1 × 10^6^ cells/mL) and PBS (100 μL); (3) *C. albicans* CCARM 14007 (100 µL of 1 × 10^6^ cells/mL) and fluconazole (2 mg/kg); or (4) *C. albicans* CCARM 14007 (100 µL of 1 × 10^6^ cells/mL) and PS1-2 peptide (2 mg/kg); and (5) only PS1-2 peptide (2 mg/kg). The mice anesthetized by inhalation of isoflurane (5%: induction, 2%: maintenance) in pure oxygen were intratracheally instilled with *C. albicans* cells. One hour after exposure, each peptide sample was intratracheally administered at the indicated dose, and the mice were monitored for signs of morbidity for 6 days. The lungs of mice euthanized by CO_2_ inhalation were excised and fixed in 4% paraformaldehyde. The fixed tissues were dehydrated using a series of ethanol solutions (50–100%) and embedded in paraffin. Paraffin-embedded tissues were sectioned at a thickness of 5 µm (Leica microtome, Deerfield, IL, USA), and the histopathology of the lung tissues was performed using alcian blue without/with eosin, and hematoxylin and eosin (H&E) staining.

TNF-α and NO production levels in blood collected from mouse hearts were determined using mouse anti-TNF-α microplate ELISA kit (R&D Systems, Minneapolis, MN, USA) and Griess reagent kit (Invitrogen, Carlsbad, CA, USA), respectively, followed by the manufacturer’s instructions.

### 3.10. Statistical Analysis

The mean values of four independent determinations ± SD and *p*-values were calculated on Excel software (Student’s *t*-test).

## 4. Conclusions

Currently, most research regarding drug-resistance is limited to the study of antibiotic resistance in bacteria, and drug-resistant fungi are not considered a serious problem. However, the emergence of multidrug-resistant fungi was clearly alarming. Therefore, the development of next-generation drugs that do not induce drug resistance and that are effective against drug-resistant fungi is required. Here, we found that the PS1-2 peptide is active as an anti-fungal agent against the drug-resistant *C. albicans* under human physiological conditions (salts, ions, and proteases), and its antifungal activity occurs via a membranolytic action. Moreover, it can inhibit *C. albicans*-induced macrophage stimulation (reduced activity of TLR-2, and TNF-α expression), and exhibits antifungal and anti-inflammatory action in a lung injury mouse model.

## Figures and Tables

**Figure 1 antibiotics-11-01779-f001:**
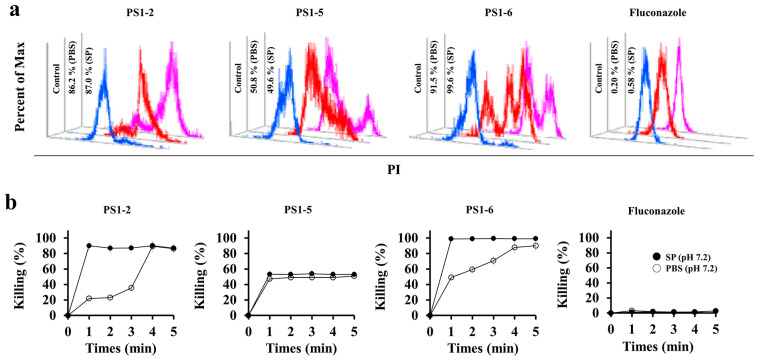
Time-kill kinetic activity of PS1-2, PS1-5, PS1-6, and fluconazole against drug-resistant *C. albicans*. (**a**,**b**) After incubation of each peptide and fluconazole in PI-pretreated *C. albicans* cells, dead cells were measured using flow cytometry. (**a**) presents the overlay spectra after 5 min incubation. (**b**) presents killing-percentages at 1 min intervals for 5 min (*n* = 4). Data are normalized to non-treated control.

**Figure 2 antibiotics-11-01779-f002:**
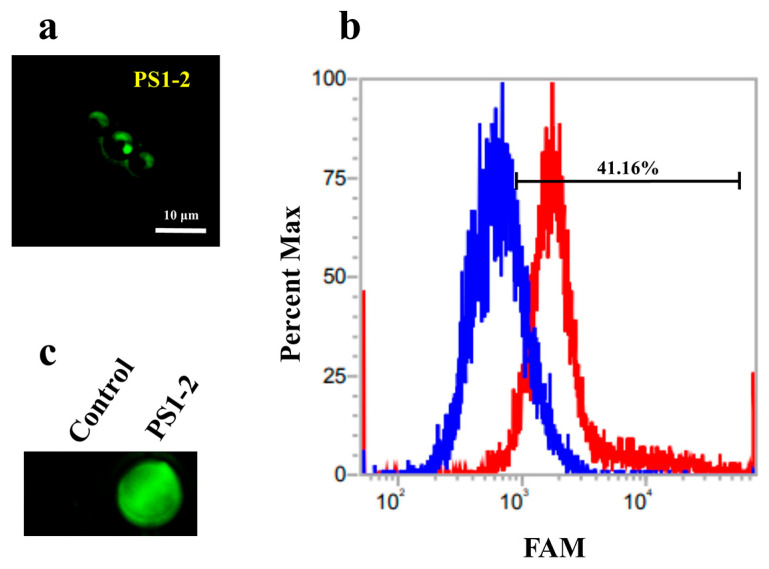
Binding affinity of 5-carboxyfluorescein (FAM)-labeled PS1-2 peptide. After incubation of FAM-labeled PS1-2 peptide for 1 min, *C. albicans* cells were three times washed with PBS, observed under fluorescence microscope (**a**), analyzed using flow cytometry (**b**), and visualized under fluorescence signal imaging system (**c**), blue line: *C. albicans*, red line: *C. albicans* with FAM-labeled PS1-2).

**Figure 3 antibiotics-11-01779-f003:**
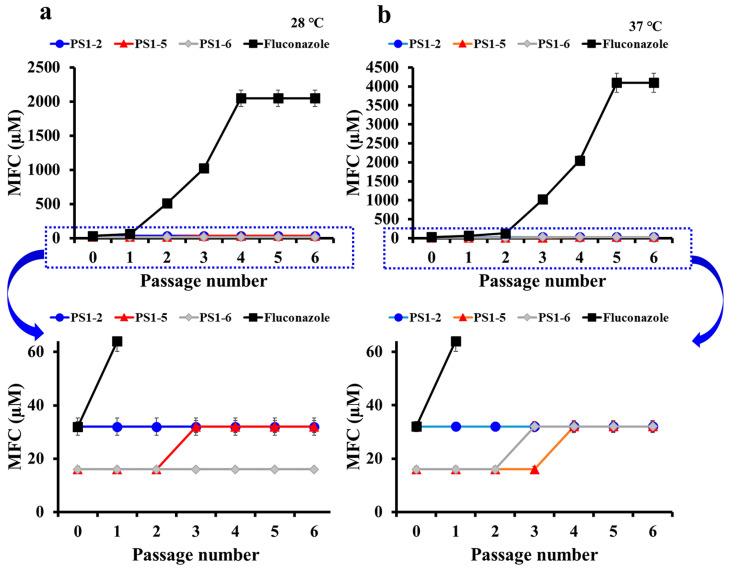
Induction of drug-resistance of peptides and fluconazole in *C. albicans* KCTC 7270 cells. Development of resistance in *C. albicans* cells were performed by the exposure to PS1-2, PS1-5, PS1-6, or fluconazole at 28 °C (**a**) or 37 °C (**b**) during 6 passages. The *Y*-axis shows the fold change of MFC from the first passage.

**Figure 4 antibiotics-11-01779-f004:**
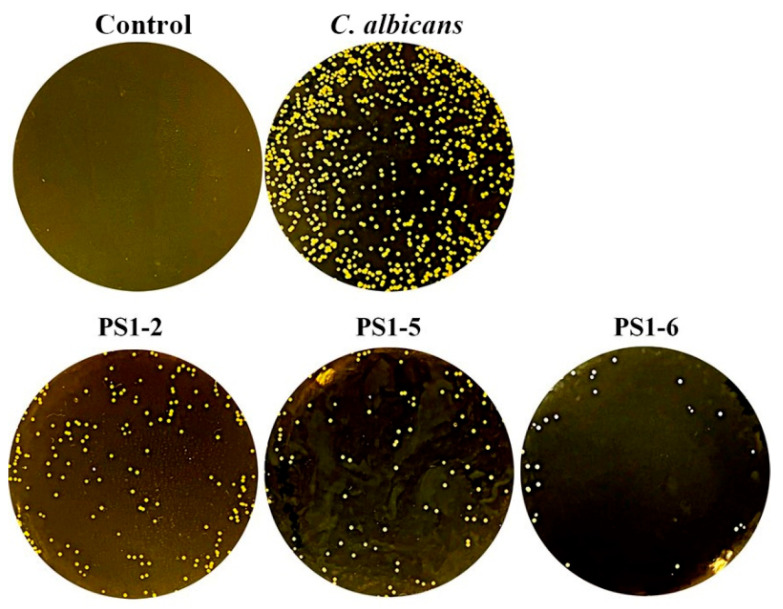
Protease protection effects of PS1-2, PS1-5, and PS1-6 peptide. After treatment of 1 µg/mL trypsin to peptide solutions of MFCs at 37 °C for 40 min, *C. albicans* CCARM 14007 was treated to the mixtures.

**Figure 5 antibiotics-11-01779-f005:**
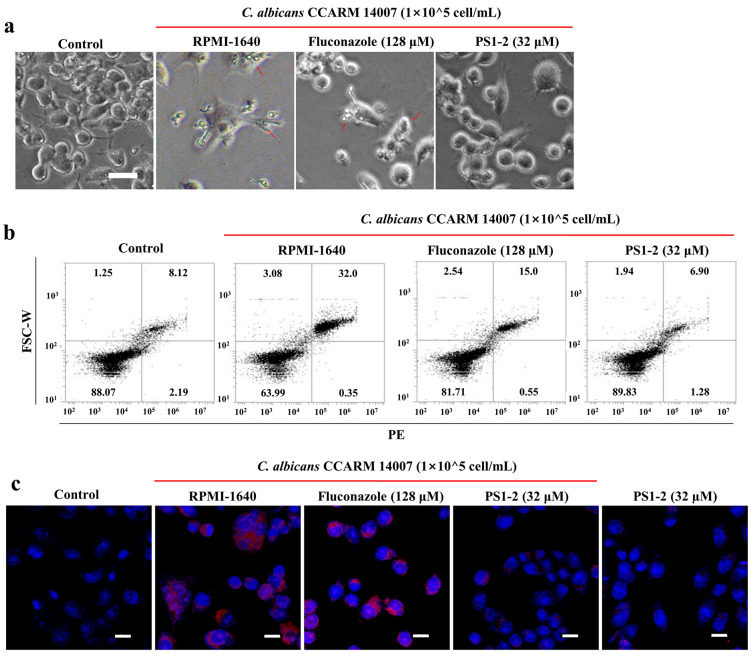
Anti-inflammatory effects of fluconazole and PS1-2. (**a**) Pre-cultured RAW 264.7 cells were stimulated with *C. albicans* CCARM 14007 in the presence or absence of fluconazole (128 µM) and PS1-2 (32 µM), followed by 24 h incubation. Arrow indicates phagocytosis. (**b**) TLR-2 expression on the surface of RAW 264.7 cells was analyzed using flow cytometry. (**c**) TNF-α over-expression in RAW 264.7 cells was visualized under confocal laser scanning microscope (CLSM). Size bar is 20 µm (**a**) and 10 µm (**c**).

**Figure 6 antibiotics-11-01779-f006:**
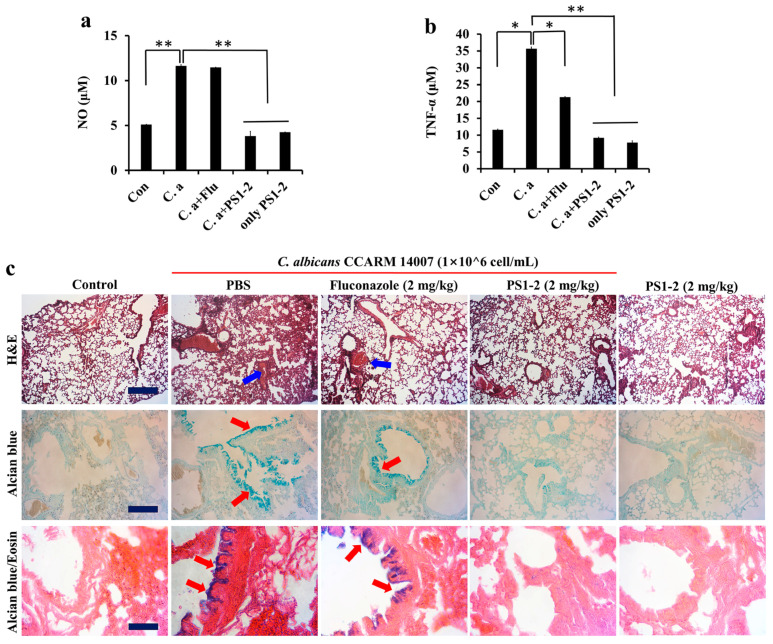
In vivo anti-inflammatory effects of peptides in lung injury. *C. albicans* CCARM 14007 (1 × 10^5^ cells) was intratracheally instilled in mice lung. One hour after instillation, peptides were intratracheally subjected, followed by monitoring for 6 days. Measurement of NO production (**a**) and TNF-α secretion (**b**) levels with whole blood serum in the presence or absence of fluconazole and PS1-2 at the indicated doses (* *p* < 0.05, ** *p* < 0.01). (**c**) Hematoxylin and eosin (H&E), alcian blue, and alcian blue and eosin staining showing induced inflammation on the lung tissues. Red arrows: an increase in the release of mucins, blue arrows: inflammatory cell infiltrates around the bronchus and alveoli. Size bar = 2 mm.

**Table 1 antibiotics-11-01779-t001:** Candidacidal activity of PS1-2, PS1-5, PS1-6 peptide, and fluconazole against *Candida albicans* CCARM 14007 (fluconazole-resistant strain) in various buffer conditions.

*C. albicans* CCARM 14007		MFC^a^ (μM)			
		PS1-2	PS1-5	PS1-6	Fluconazole
**Temperature**					
28 °C	pH				
	pH 5.5 ^a^	32	16	16	>128
	pH 6.0 ^a^	32	16	16	>128
	pH 7.2 ^a^	32	32	16	>128
	Ion (pH 5.5)				
	150 mM NaCl	32	64	32	>128
	24 mM KCl	32	64	32	>128
	6 mM CaCl_2_	64	32	64	>128
	6 mM MgCl_2_	32	32	32	>128
37 °C	pH				
	pH 5.5 ^a^	32	16	16	>128
	pH 6.0 ^a^	32	16	16	>128
	pH 7.2 ^a^	32	32	16	>128
	Ion (pH 5.5)				
	150 mM NaCl	32	64	32	>128
	24 mM KCl	32	64	32	>128
	6 mM CaCl_2_	64	32	64	>128
	6 mM MgCl_2_	32	32	32	>128

^a^ MFC is a minimum fungicidal concentration. Antifungal activity was tested in 10 mM sodium phosphate (SP) buffer containing 10% media (low ionic strength buffer).

## Data Availability

Not applicable.

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
