# Peer review of "Antifungal and Anti-Inflammatory Activities of PS1-2 Peptide against Fluconazole-Resistant Candida albicans"

_antibiotics, 2022, doi:10.3390/antibiotics11121779_

Round 1

Reviewer 1 Report

This article presented the work of the synthesized peptides in the inhibition of drug-resistant C.albicans. The work found that the PS1-2 peptide had antifungal activity under physiological conditions. This peptide caused membrane lysis and reduced the inflammatory cytokine in C.albicans-treated macrophage. It is interesting to discover novel agents that combat drug-resistant fungi. I recommend it for publication in Antibiotics after minor revision. There are some comments as follows;

1. Line 152; the authors should give more discussion related to the modification of peptides in PS1-2, PS1-5, and PS1-6 for the MFC activity. For example, its physicochemical property.

2. Line 157; give the discussion of the effect of PBS and SP buffers on the time-killing kinetics.

3. Legend of Figure 1A; give the detail of the incubated time used in the flow cytometer.

4. Figure 1B; give the error bar in the graph and detail of pH in PBS and SP buffer.

5. Line 173; It seemed that PS1-6 had the highest potency. The author should explain why they used PS1-2 in further experiments.

6. Line 178; give more discussion on the result obtained from Figures 2B and 2C.

7. Line 185; give the full name of FOBI and FACS.

8. Line 191; give the p-value to explain the significance.

9. Line 198; the author changed the tested strain to C.albicans KCTC 7270. Giving the explanation will be useful for the readers to understand the work.

10. Line 201; correct the word “first”.

11. Line 223; explain what is the morphological change.

12. Line 238; Is it (C) TNF-a expression?

13. Figure 5A and 5C; make the size bars in the figures clear.

14. Figure 6A and 6B; do statistical analysis to find out that the peptide could reduce NO and TNF-a production.

15. Line 299; add the name of the indicated buffer. Did it include 10% (line 142) or 20% of the medium?

16. Line 310 – 313; give more conditions used in the fluorescence experiment, such as excited and emitted wavelength.

17. Line 315; give the strain of C.albicans used in this experiment.

18. Line 323; rearrange the concentrations from 1 and 128 mcM.

19. Line 330; give the detail of positive and negative controls.

20. Line 333; what is an antibiotic?

21. Line 351; did animal group 3 and 4 include C.albicans?

Author Response

This article presented the work of the synthesized peptides in the inhibition of drug-resistant C.albicans. The work found that the PS1-2 peptide had antifungal activity under physiological conditions. This peptide caused membrane lysis and reduced the inflammatory cytokine in C.albicans-treated macrophage. It is interesting to discover novel agents that combat drug-resistant fungi. I recommend it for publication in Antibiotics after minor revision. There are some comments as follows;

  1. We are very grateful for the reviewer's helpful comments on this manuscript.

  1. Line 152; the authors should give more discussion related to the modification of peptides in PS1-2, PS1-5, and PS1-6 for the MFC activity. For example, its physicochemical property.
  2. A) We added some discussion.
  3. Line 157; give the discussion of the effect of PBS and SP buffers on the time-killing kinetics.
  4. A) We added some discussion.
  5. Legend of Figure 1A; give the detail of the incubated time used in the flow cytometer.
  6. A) We added the indications of (a) and (b).
  7. Figure 1B; give the error bar in the graph and detail of pH in PBS and SP buffer.
  8. A) Figure 1B was corrected according to reviewer’s comment.
  9. Line 173; It seemed that PS1-6 had the highest potency. The author should explain why they used PS1-2 in further experiments.
  10. A) We added following description “Although PS1-6 exhibits high antifungal activity as shown in Table 1 and Figure 1, previous studies have reported that it has significant toxicity [23]. Therefore, the subsequent experiments were performed with PS1-2.”
  11. Line 178; give more discussion on the result obtained from Figures 2B and 2C.
  12. A) We added our suggestions.
  13. Line 185; give the full name of FOBI and FACS.
  14. A) We corrected.
  15. Line 191; give the p-value to explain the significance.
  16. A) We believe that this significance does not need to be statistically explained by the P-value.
  17. Line 198; the author changed the tested strain to C.albicansKCTC 7270. Giving the explanation will be useful for the readers to understand the work.
  18. A) We measured each MFC against drug-sensitive albicans (KCTC 7270) to compare the degree and rate of drug-resistance between fluconazole and peptides.
  19. Line 201; correct the word “first”.
  20. A) We corrected.
  21. Line 223; explain what is the morphological change.
  22. A) Morphological changes are Lamellipodia extension and spreading of cells.
  23. Line 238; Is it (C) TNF-a expression?
  24. A) TNF-α over-expression in RAW 264.7 cells
  25. Figure 5A and 5C; make the size bars in the figures clear.
  26. A) We changed Figure 5.
  27. Figure 6A and 6B; do statistical analysis to find out that the peptide could reduce NO and TNF-a production.
  28. A) We added statistical analysis in Figure.
  29. Line 299; add the name of the indicated buffer. Did it include 10% (line 142) or 20% of the medium?
  30. A) We added buffer compositions. 10% medium in Line 142 is final concentration. 20% medium is right because peptides were diluted with only buffer.
  31. Line 310 – 313; give more conditions used in the fluorescence experiment, such as excited and emitted wavelength.
  32. A) The instruments used in these experiments do not specify excited and emitted wavelengths, but record or measure by fixed filters.
  33. Line 315; give the strain of C. albicans used in this experiment.
  34. A) We added strain name.
  35. Line 323; rearrange the concentrations from 1 and 128 mcM.
  36. A) We corrected.
  37. Line 330; give the detail of positive and negative controls.
  38. A) We added.
  39. Line 333; what is an antibiotic?
  40. A) We added antibiotic names.
  41. Line 351; did animal group 3 and 4 include C.albicans?
  42. A) We corrected this information.

Reviewer 2 Report

The article by Lee et al. describes three novel antifungal peptides in terms of their antifungal and anti-inflammatory efficiency and stability using a combination of in vitro and in vivo models. The authors suggest that Candida albicans does not develop resistance to PS1 peptides and remains effective in an in vitro mouse infection model. While the paper offers an important non-traditional approach to battle resistant fungal infections, there are numerous major concerns that need to be addressed before publication, including significant grammatical errors:

Major comments:

·       Lines 53-65: This paragraph is confusing in terms of talking about autoimmunity and then ending with “prevent excessive inflammatory responses” when the authors made it clear that these responses are not present in immunocompromised patients.

·       Throughout the paper, the discussion of biofilms is discussed but is not specified that it is fungal biofilms. In line 78-82, both bacteria and fungus is discussed in the context of biofilms but does not specify what type of biofilms.

·       Line 96 discusses that there are 3 repeated motifs, however, this was not explained in the introduction. Is this a characteristic of all peptides?

 For Table 1, it is said that PS1-6 “had increased sensitivity to ion changes,” but this does not appear to be the case according to the table. In fact, under different ion conditions, PS1-2 has the same MFC as PS1-6. Why was PS1-2 chosen over PS1-6 for future experiments, especially when it was shown later to be more potent towards C. albicans and more resistant to trypsin degradation?

·       Figure 1a: color legends are not defined. X axis is missing. Need a better y-axis legend, the percentage of max is confusing. Max of what?

·       Figure 1: Results were compared between tested PS1 peptides without discussing significance.

·       Figure 1B: how did the authors quantify the percent killing for each of the treatment conditions?

·       In Figure 1B, were all these samples prepared and analyzed at the same time? If so, 1-minute intervals between readings for each of these treatments seems difficult and no mention was made with regards to the number of replicates were made for each condition.

Figure 2, like Figure 1, needs additional information to guide readers through the significance of the figure. This would increase clarity when looking at the results. How many times were cells washed? What were they washed with? Methods are lacking. What strain of C. albicans cells were used?

·       Figure 2 needs a more detailed legend, especially for panel (b). What are the colors? What is shown in panel (b) – missing a scale bar?

·       In Figure 2b, the 41.16% should be explained in the text – what is this showing?

·       Line 195: states the data not shown, but the data are shown in Fig 3. Is that correct?

For Figure 3, a different strain of C. albicans cells (KCTC 7270) was used in this experiment. Why?

·       Why are the initial MFCs obtained using a different strains of C. albicans compared to follow-up results (ex. Fig 3)? The drug-resistance experiments need to be repeated in CCARM 14007 strain

·       Please use Image J to quantify foci in Fig 4 and carry out statistical analysis

·       The investigated PSI-2 peptide was not discussed in the introduced section beside the methods.

·       Morphological changes in macrophages were not described sufficiently. Authors should carefully reference figures and describe the morphological changes

·       Explanation of choice of mouse tissue is lacking. Why lung?

o   “C. albicans is a common commensal fungus that colonizes the human skin and inside 97 the body, such as the mouth, gut, throat, and vagina [26]. It causes fungal infections in the 98 bloodstream or internal organs, including the kidneys, heart, and brain [26]”

Error in Figure 5 legend. Part C is incorrectly listed as part B. Were cytotoxicity assays performed with the PS1 peptides and macrophages? I recommend adding this information to accompany the morphological images.

·       What is the medium in control no yeast? Why is yeast control labeled RPMI? Figure 5B does not show any inhibition. Is control in Fig 5b also RPMI1640?

For Figure 6a and 6b, are these results significant? There does not appear to be any analyses of the data besides what is shown. The authors need to include statistical significance. Figure 6a has a difference of a few uM between treatments, for example.

The methods section lacks critical information about the experiments. For example, like mentioned above, how many times and with what solution were cells washed? I recommend writing methods in more detail for reproducibility.

·       Line 218: what were the concentrations used to pre-treat RAW macrophages? What does pre-culture mean?

·       Biological replicates were not discussed in any experiments included in this article.

·       Headings throughout the paper should be more specific and include results drawn from the experiments

·       There needs to be more discussion in the results section. The results are stated clearly but the paper needs to state the significance of the results more clearly.

·       In the protease degradation activity, is it shown that an increase in hydrophobicity of PS1-6 is sterically hindered? If so, this should be explained more. If not, this needs to move to the discussion section and more detail included.

·       Physiological relevance of monovalent and divalent ionic buffers should be discussed.

·       “Morphological alterations associated with macrophage activation, and phagocytosis were observed in the presence of C. albicans cells as well as in the presence of C. albicans cells treated with fluconazole” What kind of changes? This should be validated using qPCR, ELISA (cytokine production), etc

·       Figure 6C is not very clear on what is supposed to be shown in the images. Needs further elaboration on the visuals and possibly include arrows on images to point out specific positions that are significant.

·       Can the authors comment on whether the FAM-label at the N-terminus of PS1-2 affected the binding activity of PS1-2 to C. albicans?

·       More explanation is needed on the interpretation of the FACS results in Figure 2.

·       Did trypsin affect C. albicans survival?

·       Additionally, images plus quantified C. albicans colonies presented in bar graph form would have been easier for the viewer to understand.

·       Why is focus being drawn to PS1-6 in the protease degradation assays when the paper expressly focuses on PS1-2 in the title and the binding experiments?

·       No mention was made as to what specific morphological changes they were looking for to associate with macrophage activation. Because of this, I’m not sure what I should be interpreting from Figure 5A with the brightfield images.

·       What specific element of the flow cytometry data for TLR-2 shown in Figure 5B should be focused on? Are there replicates? Why is fluconazole effective now when it did not have any effect in the previous experiment (Fig 1b)?

·       Quantifying and normalizing the signal visualized with Figure 5C would have been a better way of expressing differences in TNF-α expression. In a similar vein, why weren’t these results validated through an ELISA performed on collected supernatant?

·       What is the relevance of using the Alcian blue stain?

·       No mention was made to the number of mice in each of these treatment conditions.

Minor comments:

·        Line 29: change “are currently” to is currently

·        Line 32: Excess spacing is present.

·        spacing between degree and C should be consistent (ex. table 1 and Fig3)

·        Line 41: “to hospitalize 75 000 patients per year” is in a different font.

·        Line 41: “75 000” should have a comma.

·        Line 42: Plurality issue

·        Line 55: remove the period before [14,15]

·        Line 58: “pathogenic fungi [15]. This recognition is mainly stimulated by unmasked β” is in a different font.

·        Line 61: ROS is not abbreviated appropriately. This abbreviation can be removed completely since it is not used again in the paper.

·        Line 71: reference missing…and a broad spectrum of activity

·        Line 86: “the” not italicized

·        Line 93: space missing between and-eliminating

·        Line 96: remove comma after “peptides,”

·        Liner 101-103: dental foci, gastric lumen, etc. are not organs. Pleas rewrite this sentence and add cystic fibrosis and asthma patients

·        Table 1 formatting needs to be addressed. Something is wrong with column 1

·        how long is the recovery period for each sample in Table 1

·        Line 162: text should refer to Figure 1A

·        Figure 1B: figure legend should state that the data are normalized to non-treated control

·        What is FAM, FOBI, CLMS, ALI, FACS, HL2-H

·        Figure 2 legend lacks details. What is 2b? color coding, method, etc.

·        What is the rationale to follow up on PS1-2 when PS1-6 was the most effective AMP

·        Line 187-88: please re-write the sentence

·        Line 192: antivirals, not antibiotics

·        Line 201: “firs” should be spelled as “first”

·        Figure 3: the scale needs to be changed for PS1 peptides. Also, are the cells even alive?

·        Fig 3 legend: spell out numbers <10 here and across the whole manuscript (not time)

·        Line 217-18: please re-write the first sentence. It sounds like the authors used one cell

·        Figure 5a: What are the arrows pointing to?

·        Figure 5C: it is difficult to see the red staining. Perhaps the color should be changed to pseudo color that is more distinct (ex. yellow).

·        Figure 4 legend two last sentences are redundant

·        Line 231: Why is TNF-alpha being defined after its initial introduction?

·        Line 333: “Cells” should not be capitalized.

·        Line 339: What rpm/ xg were the samples spun down at?

·        Line 355: What method was used to administer the peptides?

·        Line 371: Why is discussing C. auris significant? The first time it was mentioned in the article was within the conclusion

Author Response

The article by Lee et al. describes three novel antifungal peptides in terms of their antifungal and anti-inflammatory efficiency and stability using a combination of in vitro and in vivo models. The authors suggest that Candida albicans does not develop resistance to PS1 peptides and remains effective in an in vitro mouse infection model. While the paper offers an important non-traditional approach to battle resistant fungal infections, there are numerous major concerns that need to be addressed before publication, including significant grammatical errors:

Major comments:

  • Lines 53-65: This paragraph is confusing in terms of talking about autoimmunity and then ending with “prevent excessive inflammatory responses” when the authors made it clear that these responses are not present in immunocompromised patients.
  1. This paragraph was modified.

  • Throughout the paper, the discussion of biofilms is discussed but is not specified that it is fungal biofilms. In line 78-82, both bacteria and fungus is discussed in the context of biofilms but does not specify what type of biofilms.
  1. We added bacteria and fungus names.

  • Line 96 discusses that there are 3 repeated motifs, however, this was not explained in the introduction. Is this a characteristic of all peptides?
  1. Three motifs mean (KWYK), (RWYR), and (KWLK).

 For Table 1, it is said that PS1-6 “had increased sensitivity to ion changes,” but this does not appear to be the case according to the table. In fact, under different ion conditions, PS1-2 has the same MFC as PS1-6. Why was PS1-2 chosen over PS1-6 for future experiments, especially when it was shown later to be more potent towards C. albicans and more resistant to trypsin degradation?

  1. We described the changes of MFCs for each peptide, not the MFCs comparison of them by changing pH and ions. Although PS1-6 exhibits high antifungal activity as shown in Table 1 and Figure 1, previous studies have reported that it has significant toxicity [23]. Therefore, the subsequent experiments were performed with PS1-2.

  • Figure 1a: color legends are not defined. X axis is missing. Need a better y-axis legend, the percentage of max is confusing. Max of what?
  1. Color legends are already included in the image. The X axis is also marked to BL2-H in the image. Percent of Max is a represent way used when various data overlay on one graph, and is expressed based on the maximum peak of the cell number.

  • Figure 1: Results were compared between tested PS1 peptides without discussing significance.
  1. We added some discussions.

  • Figure 1B: how did the authors quantify the percent killing for each of the treatment conditions?
  1. As described above, killing was quantified by the number of cells uptaken by the PI dye, using flow cytometry.

  • In Figure 1B, were all these samples prepared and analyzed at the same time? If so, 1-minute intervals between readings for each of these treatments seems difficult and no mention was made with regards to the number of replicates were made for each condition.
  1. How are samples prepared at the same time? We prepared each sample at 30-minute intervals in same conditions. The error bar and replication number presented in figure and its legend.

Figure 2, like Figure 1, needs additional information to guide readers through the significance of the figure. This would increase clarity when looking at the results. How many times were cells washed? What were they washed with? Methods are lacking. What strain of C. albicans cells were used?

  1. More discussions added in result section and method was described in material and method section.

  • Figure 2 needs a more detailed legend, especially for panel (b). What are the colors? What is shown in panel (b) – missing a scale bar?
  1. We corrected figure legend. This data does not require a scale bar.

  • In Figure 2b, the 41.16% should be explained in the text – what is this showing?
  1. We added.

  • Line 195: states the data not shown, but the data are shown in Fig 3. Is that correct?
  1. “Data not shown” is correct. Figure 3 presented during six passages, but this description is over six passages.

  • For Figure 3, a different strain of C. albicans cells (KCTC 7270) was used in this experiment. Why? Why are the initial MFCs obtained using a different strains of C. albicans compared to follow-up results (ex. Fig 3)? The drug-resistance experiments need to be repeated in CCARM 14007 strain
  1. We partially agree with the reviewer's opinion. However, in order to show the difference with conventional antibiotics, we measured each MFC against drug-sensitive albicans (KCTC 7270).

  • Please use Image J to quantify foci in Fig 4 and carry out statistical analysis
  1. We used the images to visualize the colonies and do not believe that this result requires statistical analysis.

  • The investigated PSI-2 peptide was not discussed in the introduced section beside the methods.
  1. We added sequence of PS1-2 in introduction section and this is already described in 2.1 result section.

  • Morphological changes in macrophages were not described sufficiently. Authors should carefully reference figures and describe the morphological changes
  1. We added “Lamellipodia extension and spreading of cells~”

  • Explanation of choice of mouse tissue is lacking. Why lung?

“C. albicans is a common commensal fungus that colonizes the human skin and inside 97 the body, such as the mouth, gut, throat, and vagina [26]. It causes fungal infections in the 98 bloodstream or internal organs, including the kidneys, heart, and brain [26]”

  1. We have already mentioned that albicans was injected intratracheally. We tried to confirm anti-inflammatory effects of peptides in an acute lung injury model, which is the most problematic in infectious diseases. 

Error in Figure 5 legend. Part C is incorrectly listed as part B. Were cytotoxicity assays performed with the PS1 peptides and macrophages? I recommend adding this information to accompany the morphological images.

  1. We corrected mistakes. We did not perform cytotoxicity for macrophage because these data have a group treated with only peptides.

  • What is the medium in control no yeast? Why is yeast control labeled RPMI? Figure 5B does not show any inhibition. Is control in Fig 5b also RPMI1640?
  1. Please look carefully at the figure. It is divided into treated and untreated albicans with a red line. Control is only Raw264.7 cells and RPMI-1640 is Raw 264.7 cells treated with C. albicans in the absence of peptide or fluconazole.

For Figure 6a and 6b, are these results significant? There does not appear to be any analyses of the data besides what is shown. The authors need to include statistical significance. Figure 6a has a difference of a few uM between treatments, for example.

  1. We added Statistical analysis.

The methods section lacks critical information about the experiments. For example, like mentioned above, how many times and with what solution were cells washed? I recommend writing methods in more detail for reproducibility.

  1. Some methods were corrected.

  • Line 218: what were the concentrations used to pre-treat RAW macrophages? What does pre-culture mean?
  1. This information was described in method section. After seeding Raw264.7 cells, the plate was incubated for 24 h. So, we expressed to “pre-cultured”

  • Biological replicates were not discussed in any experiments included in this article.
  1. In the statistical analysis, it was expressed as "four independent".

  • Headings throughout the paper should be more specific and include results drawn from the experiments
  1. The title of the paper cannot be edited at this time.

  • There needs to be more discussion in the results section. The results are stated clearly but the paper needs to state the significance of the results more clearly.
  1. We agree with the reviewer's comments, and some have been added or modified.

  • In the protease degradation activity, is it shown that an increase in hydrophobicity of PS1-6 is sterically hindered? If so, this should be explained more. If not, this needs to move to the discussion section and more detail included.
  1. The increase in hydrophobicity induces self-aggregation of the peptides, and these aggregates cannot be digested by trypsin because the cleavage sites of the peptides are not exposed. Peptides composed of D-amino acids or peptoids cannot be easily cleaved because proteolytic enzymes do not recognize the cleavage site.

  • Physiological relevance of monovalent and divalent ionic buffers should be discussed.
  1. We believe that importance of ions and maintenance of antifungal activity of peptides have been sufficiently discussed. Specific physiological environments in various tissues and organs in the body cannot be described because it is not known where albicans infection may occur.

  • “Morphological alterations associated with macrophage activation, and phagocytosis were observed in the presence of C. albicans cells as well as in the presence of C. albicans cells treated with fluconazole” What kind of changes? This should be validated using qPCR, ELISA (cytokine production), etc
  1. We agree and appreciate the reviewer's helpful comments. However, we focused on visualization and it is difficult to get results within the revision period. We will perform several experiments on different projects, according to the reviewer's opinion.

  • Figure 6C is not very clear on what is supposed to be shown in the images. Needs further elaboration on the visuals and possibly include arrows on images to point out specific positions that are significant.
  1. Figure 6 was modified.

  • Can the authors comment on whether the FAM-label at the N-terminus of PS1-2 affected the binding activity of PS1-2 to C. albicans?
  1. Since the FAM labeling on PS1-2 increases its hydrophobicity, this may affect its antifungal activity and mechanism against albicans. Therefore, we treated C. albicans with a mixture of FAM-labeled PS1-2 and free PS1-2 at a 1:9 molar ratio to minimize this effect.

  • More explanation is needed on the interpretation of the FACS results in Figure 2.
  1. We added more description.

  • Did trypsin affect C. albicanssurvival?
  1. Trypsin does not affect albicans survival.

  • Additionally, images plus quantified C. albicanscolonies presented in bar graph form would have been easier for the viewer to understand.
  1. We agree with the reviewer's opinion, but we used the images to visualize the colonies.

  • Why is focus being drawn to PS1-6 in the protease degradation assays when the paper expressly focuses on PS1-2 in the title and the binding experiments?
  1. We just described the results, we didn't focus. We added some discussions.

  • No mention was made as to what specific morphological changes they were looking for to associate with macrophage activation. Because of this, I’m not sure what I should be interpreting from Figure 5A with the brightfield images.
  1. We have added an explanation to the text.

  • What specific element of the flow cytometry data for TLR-2 shown in Figure 5B should be focused on? Are there replicates? Why is fluconazole effective now when it did not have any effect in the previous experiment (Fig 1b)?
  1. The sum of the values in the first and fourth quadrants represents the percentage of cells in which TLR-2 is expressed on the Raw 264.7 cell surface. The experiment was repeated 4 times, but the results showed a similar pattern. Whether it is the effect of fluconazole itself, in fact, we difficult to describe this result.

  • Quantifying and normalizing the signal visualized with Figure 5C would have been a better way of expressing differences in TNF-α expression. In a similar vein, why weren’t these results validated through an ELISA performed on collected supernatant?
  1. In the opinion of the reviewer, it is more desirable to compare with quantified figures. We focused on visualization in vitro We tried to show the over-expression of TNF-α by in vivo results.

  • What is the relevance of using the Alcian blue stain?
  1. In particular, alcian blue was used to detect mucins secreted from the lungs undergoing severe inflammation.

  • No mention was made to the number of mice in each of these treatment conditions.
  1. We use 4 mice per group.

Minor comments:

  • Line 29: change “are currently” to is currently
  1. “are currently” is right.
  • Line 32: Excess spacing is present.
  1. This is the format of the journal and will be edited by the editorial office.
  • spacing between degree and C should be consistent (ex. table 1 and Fig3)
  1. We corrected.
  • Line 41: “to hospitalize 75 000 patients per year” is in a different font.
  1. This sentence is same font to others in our Word version.
  • Line 41: “75 000” should have a comma.
  1. We added comma.
  • Line 42: Plurality issue
  1. No problem
  • Line 55: remove the period before [14,15]
  1. We deleted.
  • Line 58: “pathogenic fungi [15]. This recognition is mainly stimulated by unmasked β” is in a different font.
  1. This sentence is same font to others in our Word version.
  • Line 61: ROS is not abbreviated appropriately. This abbreviation can be removed completely since it is not used again in the paper.
  1. “ROS” is removed.
  • Line 71: reference missing…and a broad spectrum of activity
  1. We added two references.
  • Line 86: “the” not italicized
  1. We corrected.
  • Line 93: space missing between and-eliminating
  1. Space is added.
  • Line 96: remove comma after “peptides,”
  1. We removed comma.
  • Liner 101-103: dental foci, gastric lumen, etc. are not organs. Pleas rewrite this sentence and add cystic fibrosis and asthma patients
  1. “Some organ—“ is removed.
  • Table 1 formatting needs to be addressed. Something is wrong with column 1
  1. This will be edited by the editorial office
  • how long is the recovery period for each sample in Table 1
  1. Sorry, we don't understand the question.
  • Line 162: text should refer to Figure 1A
  1. “Figure 1” is right.
  • Figure 1B: figure legend should state that the data are normalized to non-treated control
  1. We added “Data are normalized to non-treated control.”
  • What is FAM, FOBI, CLMS, ALI, FACS, HL2-H
  1. All added full names or description.
  • Figure 2 legend lacks details. What is 2b? color coding, method, etc.
  1. We corrected.
  • What is the rationale to follow up on PS1-2 when PS1-6 was the most effective AMP
  1. We added the reason
  • Line 187-88: please re-write the sentence
  1. We corrected.
  • Line 192: antivirals, not antibiotics
  1. Do you mean "Antifungals"?
  • Line 201: “firs” should be spelled as “first”
  1. We corrected.
  • Figure 3: the scale needs to be changed for PS1 peptides. Also, are the cells even alive?
  1. We changed figure 3. In MFC, albicans is mostly killed, and half of the cells are viable at sub-lethal concentration,

        Fig 3 legend: spell out numbers <10 here and across the whole manuscript (not time)

  1. We don’t understand this comment.
  • Line 217-18: please re-write the first sentence. It sounds like the authors used one cell
  1. We corrected.
  • Figure 5a: What are the arrows pointing to?
  1. Arrow indicates phagocytosis.
  • Figure 5C: it is difficult to see the red staining. Perhaps the color should be changed to pseudo color that is more distinct (ex. yellow).
  1. We changed bar color.
  • Figure 4 legend two last sentences are redundant
  1. The redundant sentences was removed.
  • Line 231: Why is TNF-alpha being defined after its initial introduction?
  1. Sorry, we don't understand the question.
  • Line 333: “Cells” should not be capitalized.
  1. We corrected.
  • Line 339: What rpm/ xg were the samples spun down at?
  1. At 1,000×g
  • Line 355: What method was used to administer the peptides?
  1. “intratracheally ”
  • Line 371: Why is discussing C. aurissignificant? The first time it was mentioned in the article was within the conclusion
  1. It was included to emphasize the urgency of developing antifungal peptides as new antifungal agents due to the emergence of multi-drug resistant fungi. However, based on the reviewer's comment, this part was deleted.

Round 2

Reviewer 2 Report

Thank you for addressing the initial comments. Here are some comments that were either not addressed or the responses were not clear:

Thank you for providing the statistical significance in Fig6a and b; however, it is not clear how the data were analyzed. Is the Ca+Flu analyzed compared to Ca alone? If so, then how is p<0.05 if the bars are exactly the same. The same applies to Fig6b.

For Figure 6a and 6b, are these results significant? There does not appear to be any analyses of the data besides what is shown. The authors need to include statistical significance. Figure 6a has a difference of a few uM between treatments, for example.

  1. We added Statistical analysis.

The authors did not address the following comment that referred to result headings, not to the paper title. By the way, the title can be edited before the final proof.

  • Headings throughout the paper should be more specific and include results drawn from the experiments
  1. The title of the paper cannot be edited at this time.

Numbers (<10) should be written out. For example, line 351: 6 times, should read six times.

Fig6 missing legend details. What are the arrows pointing to? It is explained in the text but should also be included in the figure legend.

The authors should add the rationale provided for using the Alcian blue stain.

  • What is the relevance of using the Alcian blue stain?
  1. In particular, alcian blue was used to detect mucins secreted from the lungs undergoing severe inflammation.

 Line 43: Plurality issue remains

Author Response

Thank you for addressing the initial comments. Here are some comments that were either not addressed or the responses were not clear:

Thank you for providing the statistical significance in Fig6a and b; however, it is not clear how the data were analyzed. Is the Ca+Flu analyzed compared to Ca alone? If so, then how is p<0.05 if the bars are exactly the same. The same applies to Fig6b.

For Figure 6a and 6b, are these results significant? There does not appear to be any analyses of the data besides what is shown. The authors need to include statistical significance. Figure 6a has a difference of a few uM between treatments, for example.

  1. We added Statistical analysis.
  1. We corrected.

The authors did not address the following comment that referred to result headings, not to the paper title. By the way, the title can be edited before the final proof.

  • Headings throughout the paper should be more specific and include results drawn from the experiments
  1. The title of the paper cannot be edited at this time.
  2. We're sorry you didn't understand the comment at first. We corrected the result headings.

Numbers (<10) should be written out. For example, line 351: 6 times, should read six times.

  1. We corrected.

Fig6 missing legend details. What are the arrows pointing to? It is explained in the text but should also be included in the figure legend.

  1. We added.

The authors should add the rationale provided for using the Alcian blue stain.

  • What is the relevance of using the Alcian blue stain?
  1. In particular, alcian blue was used to detect mucins secreted from the lungs undergoing severe inflammation.
  1. Alcian blue is a stain that is used to visualize acidic epithelial, and connective tissue mucins. Respiratory tracts overproduce mucus during acute injuries and in chronic conditions containing cystic fibrosis, bronchitis, and asthma [32,33].

 Line 43: Plurality issue remains

  1. We corrected.